# Reef islands have continually adjusted to environmental change over the past two millennia

P. S. Kench [1] ✉, C. Liang [2,3], M. R. Ford [3], S. D. Owen[1], M. Aslam[4], E. J. Ryan[3], T. Turner[3], E. Beetham[3], M. E. Dickson[3], W. Stephenson [5], A. Vila-Concejo [6] & R. F. McLean[7]

Global environmental change is identified as a driver of physical transformation of coral reef islands over the past half-century, and next 100 years, posing major adaptation challenges to island nations. Here we resolve whether these recent documented changes in islands are unprecedented compared with the pre-industrial era. We utilise radiometric dating, geological, and remote sensing techniques to document the dynamics of a Maldivian reef island at millennial to decadal timescales. Results show the magnitude of island change over the past half-century (±40 m movement) is not unprecedented compared with paleo-dynamic evidence that reveals large-scale changes in island dimension, shape, beach levels, as well as positional changes of ±200 m since island formation ~1,500 years ago. Results highlight the value of a multi-temporal methodological approach to gain a deeper understanding of the dynamic trajectories of reef islands, to support development of adaptation strategies at timeframes relevant to human security.

Global climatic change, sea-level rise, and changes in storm magnitude and frequency pose threats to the continued physical persistence of low-lying coral reef islands, and consequently the continued human occupation of atoll nations where reef islands provide the only habitable land[1,2]. Anthropogenically driven climate change over the past century and specifically the combined effects of sea-level rise and modification in the wave climate regime[3–5], are expected to trigger substantial changes in the physical structure of islands, including erosion, increased instability, loss of freeboard relative to sea level, and in the worst cases complete loss of islands[6,7].

Early assessments of the impacts of sea-level rise on reef islands focussed on shoreline erosion, with a number of studies speculating entire loss of habitable land[8,9]. However, more recent studies using both field-based and modelling approaches indicate a broader suite of future outcomes for small islands and their communities, in which islands will continue to persist and remain available for habitation[1,6,10]. Modelling of wave interactions with reef systems and island shorelines indicates atoll islands will be subject to an increased frequency of flooding and likely salinization of groundwater tables[1,11]. For modelling purposes, these studies assumed that the geomorphic structure of islands, including size and elevation, remain constant, whereas remote sensing[12–17] and field-based studies[18,19] have highlighted the dynamic physical characteristics of islands which can change their shape, size, location and elevation on coral reef platforms from event to multi-decadal timescales. Collectively, these studies have provided important developments in refining the tangible and immediate threats to island communities, including the likely increase in flooding hazards, and the rates, styles and magnitude of physical island change. More recent modelling studies have begun to examine the physical island response to wave and sea-level changes and show that alongshore redistribution of sediments, and wave overtopping

[1]Department of Geography, National University of Singapore, Singapore, Singapore. [2]Department of Environmental Management, Southern Institute of Technology, Invercargill, New Zealand. [3]School of Environment, University of Auckland, Auckland, New Zealand. [4]Small Island Research Station, Fares-Maathodaa Island, Huvadhoo Atoll, Maldives. [5]School of Geography, University of Otago, Dunedin, New Zealand. [6]School of Geosciences, University of Sydney, Sydney, Australia. [7]School of Science, University of New South Wales, Canberra, ACT, Australia. ✉e-mail: pkench@nus.edu.sg

and overwash sedimentation provide physical process mechanisms for observed island transformations[6,7,20,21].

The lens of existing studies has been on recent responses in the context of changing sea level, with an implicit assumption that reef islands remained relatively stable in position prior to the measured acceleration in sea-level rise over the past century. The assumption of island stability is largely untested, and it is unclear whether recent (past decades) or future changes are unusual in the context of the history of island physical dynamics. Understanding the magnitudes and trajectories of island change, and knowing which parts of islands are changing and which are stationary, is fundamental to inform robust adaptation and land-use planning in island nations. However, studies that examine contemporary island change are still few, and the temporal scale of analysis is not well calibrated to the longer-term (millennial-scale) context of island physical dynamics.

The gap in understanding medium (centennial) to long-term (millennial) island dynamics has emerged as a consequence of the temporal focus of existing island studies. Most recent studies of island dynamics have largely focussed on island change over the past half-century, the timescale most relevant to discourses of global climate change and over which the impacts of anthropogenically influenced climatic change should be observable. The magnitude and rates of change within this time window have now been identified for more than 1100 islands, mostly in the tropical Pacific[14–17,22] and to a lesser extent the Indian Ocean[23,24] (Fig. 1). Several key observations of the planform properties of islands have emerged from this large dataset. First, most islands have exhibited physical change on their reef surfaces over recent decades. Second, the dominant mode of response has been the expansion of islands on reef surfaces (>53%). Third, total loss of islands is rare (three islands, 0.3% of dataset) and, where it has occurred it has resulted primarily from storm impacts and affected the smallest of islands, that were not permanently inhabited. Fourth, contraction of islands through erosion has been observed in <34% of islands (Fig. 1a). Collectively, these studies have been unable to establish specific environmental drivers of island change, nor directly and unambiguously implicate climatic change as a mechanism for observed changes. To date, local-scale processes appear to have blurred any climatic change signal[22,24].

The multi-decadal timescale of existing island change studies has been temporally constrained by the availability of high-quality remote sensing imagery (Fig. 1a). Such a multi-decadal focus may also have served to reinforce the perception that island change is a recent phenomenon triggered by global climatic change in the Anthropocene. However, whether this behaviour over the past half-century is unprecedented in the context of the longer history of island persistence on reef surfaces has yet to be resolved. Equally critical for future adaptation considerations is whether the recent amplitude of change is extraordinary compared to the more distant past (centennial-millennial scales).

Studies of the formation of reef islands have typically focussed on evolution over millennial timeframes based on radiometric dating of island sediments that have resolved the onset, and window of island accumulation in the mid-to-late Holocene (Fig. 1b)[25–33]. Such studies have also yielded valuable information on the relationship between sea level and island formation, showing that islands have formed at different sea-level stages during the mid-to-late Holocene (Fig. 1b). The timeframe of island accumulation has also been shown to vary. For example, Boduhini and Dhakandhoo islands in the Maldives (b and h in Fig. 1b) formed across discrete 1500- and 1000-year periods[26,27]. Formation of islands in discrete phases, in response to storm processes, has also been identified in Tuvalu and the Marshall Islands (q, u and v in Fig. 1b)[28,29]. In contrast, Vaadhoo in the Maldives, and Warraber Island in the Torres Strait (g and l in Fig. 1b) have evolved continuously since their initial formation 4500 and 6000 years ago, respectively[30,31]. Such observations suggest that, for some islands, change is likely to have been an ongoing process, and for those islands that have been inhabited for the past 2000 years, communities have adapted to these changes.

Implicit in the methodological approach of evolutionary studies, is the treatment of the extant volume of island sediment, and the island footprint, as the terminal endpoint of formation, employing radiometric evidence to account for the island at the time of the study, rather than a point in the evolutionary trajectory. This approach can lead to an assumption that islands have incrementally expanded their footprint on reef surfaces over their formation and necessarily obscures any short (decadal) to medium-term (centennial) dynamism

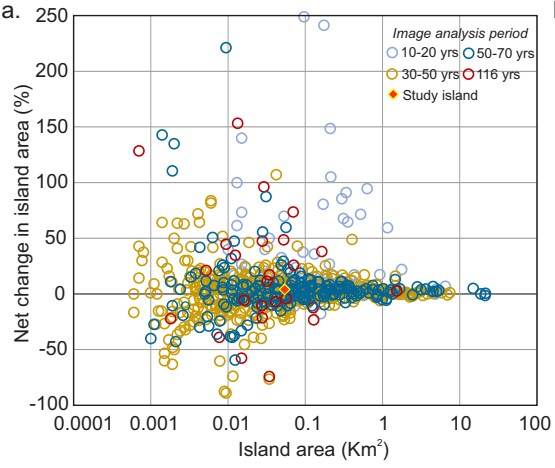

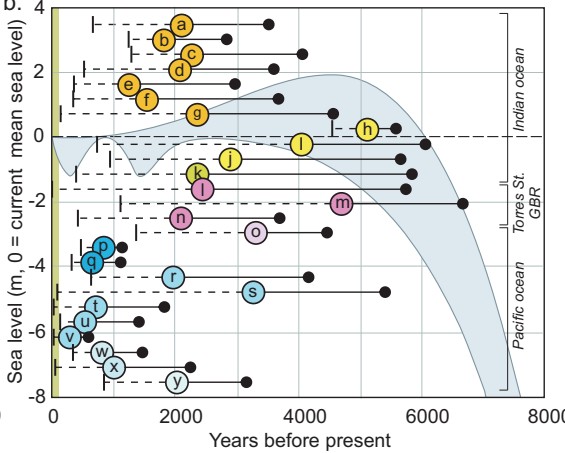

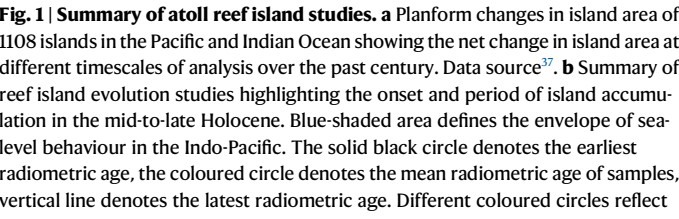

**Fig. 1 | Summary of atoll reef island studies. a** Planform changes in island area of 1108 islands in the Pacific and Indian Ocean showing the net change in island area at different timescales of analysis over the past century. Data source[37]. **b** Summary of reef island evolution studies highlighting the onset and period of island accumulation in the mid-to-late Holocene. Blue-shaded area defines the envelope of sea-level behaviour in the Indo-Pacific. The solid black circle denotes the earliest radiometric age, the coloured circle denotes the mean radiometric age of samples, vertical line denotes the latest radiometric age. Different coloured circles reflect islands in different reef provinces and the letter inside the circle denotes specific islands, a = Mainadhoo, b = Boduhini, c = Galamadhoo, d = Baavanadhoo, e = Kandahalagalaa, f = Kondey, g = Vaadhoo, h = Dhakandhoo, i = Hulhudhoo, j = Thiladhoo, k = Cocos (keeling) Isld., l = Warraber, m = Bewick, n = Lady Elliot Isld., o = Mba, p = Tepuka, q = Tutaga, r = Laura, s = Jabat, t = Jeh, u = Jabnodren, v = Jin, w = Malamala, x = Navini, y = Makin. Source references of data are listed in Supplementary Table 1.

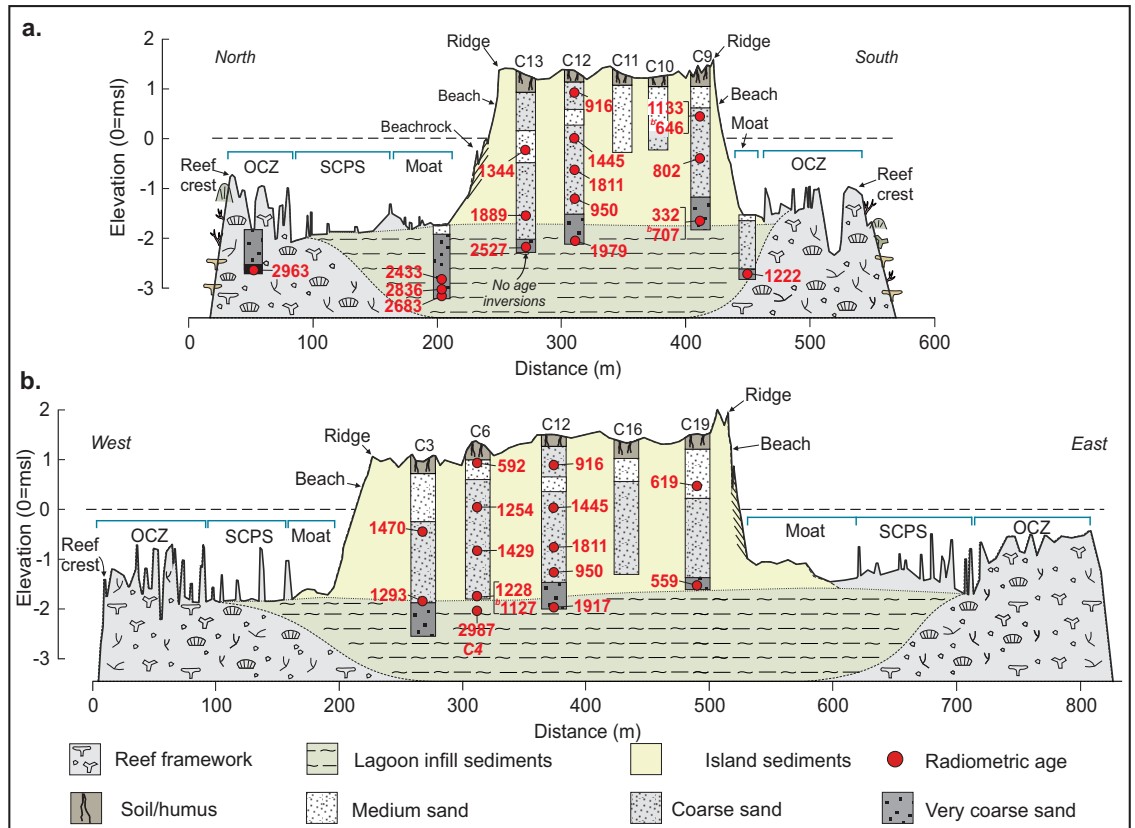

**Fig. 2 | Summary of Kandahalagalaa reef platform and island topography, and chronostratigraphy. a** North–South topographic survey along transect T3 with summary core hole stratigraphy and radiometric ages. **b** West–East topographic survey showing summary core hole stratigraphy and radiometric ages. The location of profiles is shown in Supplementary Fig. 2a. Radiometric ages represent the median age of the calibrated age range. All radiometric age details are presented in Supplementary Table 2.

in island size, shape and location. In part, these interpretations are constrained by the fact that very few studies have a sufficient number of samples (cores) or radiometric ages in vertical sequences to resolve depositional histories in detail. Furthermore, while a few studies have identified morphological features that indicate islands have occupied different positions on their reef surface[30,32], chronologies have not been established for these features to resolve the medium-term (centennial) dynamism of islands in their evolutionary history.

There have been few attempts to bridge centennial to millennial-scale understanding of island formation with short-term (decadal) observations of island dynamics, to establish whether short-term observations of change are unusual or unprecedented in the lifespan of an island. Here we combine evidence of the dynamism of an island in the Maldives, that spans geological to modern timescales drawing on a range of methodological approaches that include geological indicators (e.g., beachrock), radiometric evidence and remote sensing analysis (see "Methods"). Study site selection was determined by the availability of detailed datasets for each of these methodological approaches and timescales of consideration. Specifically, the study island has a high density of radiometric ages on island formation, detailed geomorphic and radiometrically constrained analysis of beachrock, and a high frequency of available aerial imagery. This breadth and depth of data are unprecedented among atoll islands across the Indo-Pacific. Located in Huvadhoo atoll in the southern Maldives (Supplementary Figs. 1 and 2) Kandahalagalaa island has persisted on its reef platform for the past 1500 years[33], a period in which sea level has oscillated by ±0.9 m in the region[34]. The study examines the physical dynamism of the island over the past millennia, determines whether recent changes are unusual in the context of historic change and

highlights the implications of this multi-temporal approach to inform land-use planning and formulation of adaptation strategies.

## Results

The results are structured to consider island dynamics at three time-scales; millennial and centennial timeframes to resolve historical patterns of change; and, recent changes that span the last half-century. Results are subsequently examined to address the question of whether the most recent changes are unusual in the context of past island changes.

### Millennial-scale island formation and adjustment

Stratigraphic analysis combined with radiometric evidence indicates that Kandahalagalaa Island formed over an infilled shallow lagoon as the reef was catching up to sea level in the late Holocene[35]. This is recorded by an in situ *Porites* sp. coral from 0.5 m below the outer reef dated 2963 cal yBP and from sediments in northern lagoon cores aged 2836 cal yBP to 2433 cal yBP (Fig. 2a). In contrast, basal sediments from the southern moat core (of 1222 cal yBP) signify later infill of the lagoon at this location (Fig. 2a; Supplementary Table 2). All island cores (20) terminated in unconsolidated medium-size sands (range −0.8 to 1.61 ø) and are primarily composed of coral (50–80%) with subordinate fractions of *Halimeda*, molluscs and foraminifera[33]. Basal sediment units in cores become coarser with depth (0.43–0.72 ø) with coral assuming smaller proportions (33–59%) and an increasing presence of *Halimeda* (max. 32%), molluscs (max. 20%) and foraminifera (max. 15%) that reflect a shallow lagoon provenance. Underlying the island is a basal unit comprising very coarse sands with inclusions of broken branch corals with ages ranging between 2989 cal yBP and -2000 cal yBP, which denote the terminal phase of lagoon infill beneath the

island (Fig. 2). Remaining island sediment ages ($n = 24$) range between 1889 and 332 cal yBP (Supplementary Fig. 3, Supplementary Table 2) suggesting that a major phase of sediment generation for island accumulation began around 1800 years ago and was sustained between 1500 and 500 years ago (Supplementary Fig. 3).

A notable feature of each of the dated island sediment cores is that in all but two cores (core 4 and 13) there are multiple age inversions (Fig. 2) that: (1) indicate there has been substantial reworking and mobility of island sediments; and (2) suggest the island is unlikely to have been positionally static throughout much of its existence. In contrast, core 13 the closest to the northern shoreline, does not exhibit age inversions, with a basal age of 2527 and an upper age of 1344 close to the current mean sea level (MSL). In the context of island change, three additional features of the radiometric ages are notable. First, the southern shoreline has a younger age range of sediments (1133–332 cal yBP, core 9) than the northern shoreline, and all dates in core 9 lack chronological coherence (Fig. 2a), indicating the southern portion of the island lagged the north in its depositional development and comprised reworked sediments. Second, the youngest island sediments are found toward the eastern and southern margins of the island ~600–332 years in age. Third, samples close to and above the current MSL are all less than 1400 years in age, suggesting the vegetated island may not have been established above sea level until the last millennium.

## Centennial-scale island change
Beachrock forms through the cementation of beach sediments by calcium carbonate cements in the intertidal zone and provides an indicator of past shoreline position, orientation and slope[36]. Eight distinctive beachrock outcrops occur on the northern and eastern Kandahalagalaa shoreline, providing geological markers of shoreline position over the past 1000 years (Fig. 3 and Supplementary Table 3). Each outcrop has a distinct planform orientation (strike) that differs from the contemporary shoreline, and their beach slope (dip) and exposure reveal substantive differences in shoreline position relative to the present beach.

At the eastern tip of the island, beachrock 1 extends 56 m from the 2019 shoreline, at a near-perpendicular angle to the island (Fig. 3a–c). This outcrop is up to 18 m in width, has a maximum elevation of 0.69 m MSL, in the upper intertidal range, and a slope of 11.3%. A striking feature of this deposit is that the beach slope faces toward the southeast (144°) indicating that historically the island was positioned behind and to the north of this outcrop. Beachrock outcrops 2, 4 and 5 each have a similar oblique orientation to the contemporary shoreline, and beach slopes that face toward the south-southeast (Fig. 3a, Supplementary Table 3). A further outcrop (beachrock 3) is oriented parallel to, and is exposed at, the current shoreline (Fig. 3a). The beach slope of this outcrop faces toward the south (190°), indicating this paleo-shoreline represents a period when this was the southern limit of the island, with the island being located to the north.

Toward the centre of the northern coastline, beachrock 6 (Fig. 3a) extends ~70 m across the inner moat, at an oblique angle (72.3°) to the shoreline, and has a maximum beach slope of 24%. In contrast to other beachrock outcrops along the northern shoreline, the beach slope of this outcrop faces toward the northwest (342°) and represents a period when the island was located to the south of this paleo-shoreline.

Notably, two beachrock outcrops are detached from the shoreline, located 57 m and 63 m offshore of the contemporary beach, in the inner moat, and these outcrops have a similar orientation to the existing shoreline (106° from north, Fig. 3a). The beach slopes of these outcrops face toward the north-northeast (16–17°) and are paleo-markers of the northern limit of the island shoreline. A striking feature of these deposits is that their maximum elevations are below the current intertidal range (−0.56 and −0.36 m MSL) with the mean

elevation of the deposits more than 0.75 m below MSL (-0.25 m below the intertidal range).

Radiometric ages of beachrock samples establish the temporal sequence of these paleo-shoreline markers (Supplementary Table 2, Fig. 3a). First, beachrock outcrops 1–5 are in the 1130–1440 cal yBP age range. Second, the oldest ages in this group are from beachrock 3, exposed at the island shoreline, at 1366 cal yBP from the upper unit and 1414 cal yBP in the lower unit. Third, there is a progressive decrease in the age of the shoreline oblique outcrops, with south-southeast facing beach slopes (outcrops 5, 4, 2 and 1) from beachrock 5 (1279 cal yBP) to the eastern beachrock 1 (1130 cal yBP). Fourth, three outcrops had much younger ages (Supplementary Table 2, Fig. 3a). Outcrop 6, extending at an oblique angle in the centre of the northern shoreline, and with northwest facing beach slope, was dated at 538 cal yBP. The offshore parallel beachrocks had ages of 562 cal yBP and 529 cal yBP for outcrops 7 and 8, respectively (Fig. 3a, Supplementary Table 2). Notably, the youngest date (529 cal yBP) was returned on the most seaward outcrop.

## Multi-decadal shoreline change
Over the last half-century (1969–2021) the planform area, shape and position of Kandahalagalaa has undergone considerable change, most notably substantial erosion of the northeastern shoreline and eastern tip of the island (Fig. 4). This transformation has been balanced by substantive accretion of the northwestern portion of the island, and consistent accretion along the southern shoreline. The net result of these changes has been a southward movement of the island and rotation of the island's long-axis toward the northwest (Fig. 4). Such changes define a large envelope of planform adjustments that encompass 68,199 m$^2$ (all land captured by any shoreline) which is 26% larger than the 1969 island footprint. The stable portion of the 1969 island outline that has not undergone change over the past half-century comprises 82%, though this area represents only 65% of the island's entire footprint over that period. Despite the substantial planform change over the 52-year period of analysis, the vegetated area of Kandahalagalaa has increased by 2360 m$^2$, or ~4.4% of the 1969 island area (Fig. 4, Supplementary Table 4). Of note, the higher frequency of sampling over the past 15 years (near annual) indicates a cyclic adjustment in the vegetated area (Fig. 4b), attributed to the time lag for colonisation of vegetation on recently accreted land.

Analysis of the 818 shoreline transects around the island[37] provides finer resolution insights on spatial variability in rates of change that contribute to the overall planform adjustments. Results show that 23.1% of transects (189) eroded, 53.7% of transects accreted, while the remaining 23.2% of transects remained relatively stable over the analysis period. Aggregated at the island scale, analysis indicates the island has a mean shoreline change envelope (SCE) of 25.2 m with a net shoreline movement (NSM) of 2.4 m over the 52 years of analysis. However, such island-averaged data masks considerable gross changes in the island footprint. Most notable is the substantial erosion of the northeastern shoreline and eastern tip of the island where there is a maximum SCE of up to 44 m, with a mean net displacement value of −34.5 m (Fig. 4a, c). At the northwest side of the island, accretion has occurred with a maximum SCE of 68.7 m (NSM of 44.3 m), and with mean values for this sector of the island being 48.2 m (SCE) and 32.6 m (NSM). Two other aspects of island shoreline change are notable. First, there has been a net shoreline movement along the southern shoreline of 9.1 m (SCE of 17.2 m). Second, there is a small sector of the central northern shoreline, in close proximity to beachrock outcrop 6, that has remained remarkably stable over the analysis period with an NSM value of less than 1 m (SCE of 5.9) (Fig. 4a, c).

## Discussion
Analysis of geomorphic, radiometric and satellite image datasets, that span the millennial to decadal timescales, each indicate

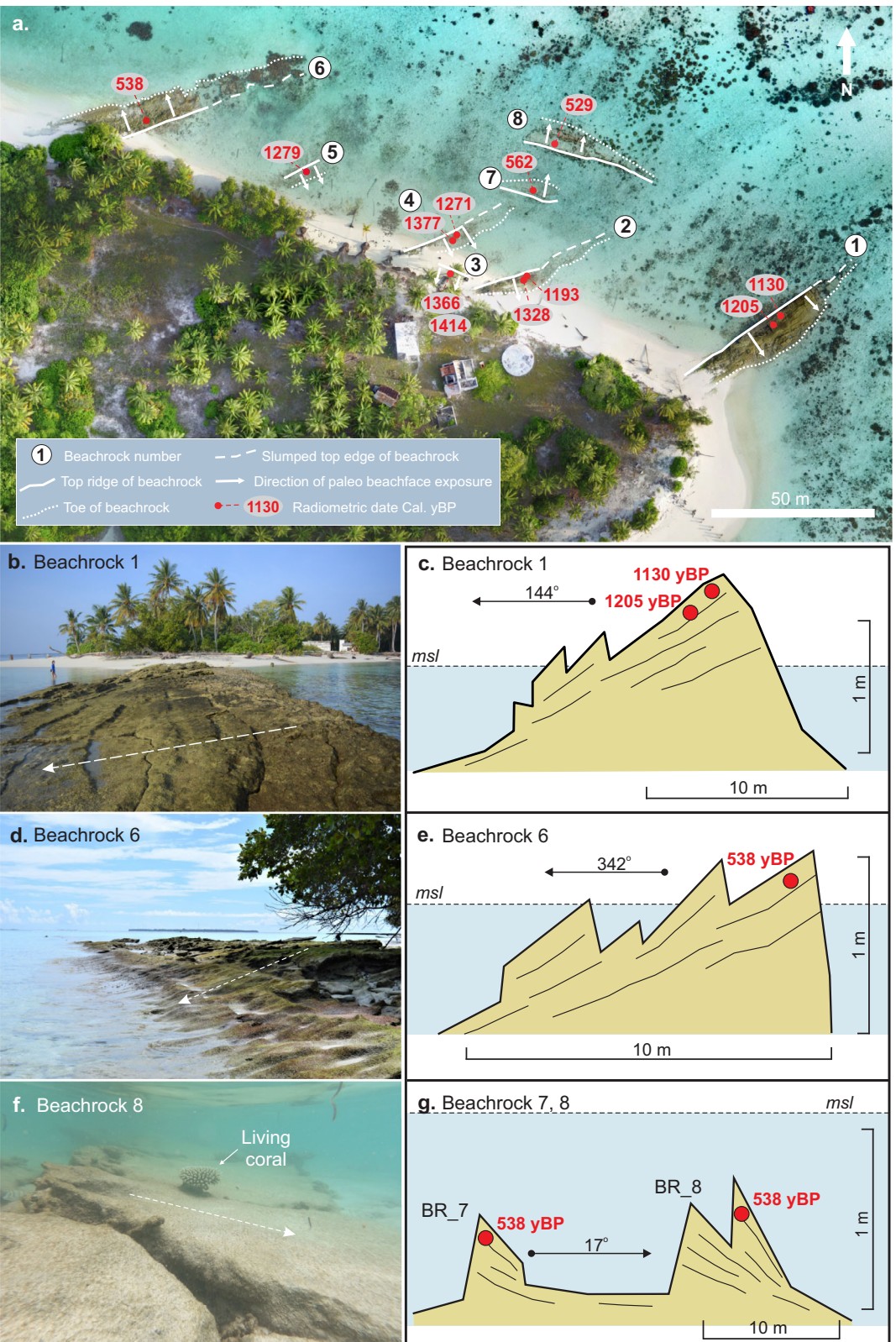

**Fig. 3 | Summary locations and characteristics of beachrock outcrops on Kandahalagalaa. a** Summary location and orientation of beachrock units. Note: arrows show the direction of exposure of beachrock slope. Source: author drone image. **b, c** Oblique photograph and survey cross-section of beachrock 1 on the southeastern end of the island. **d, e** Oblique photograph taken from the shoreline, and survey cross-section of beach rock 6. **f, g** Photograph and survey cross-section of offshore beachrock 8. Note: living coral on beachrock slope (**f**) and subtidal elevation of the outcrop (**g**).

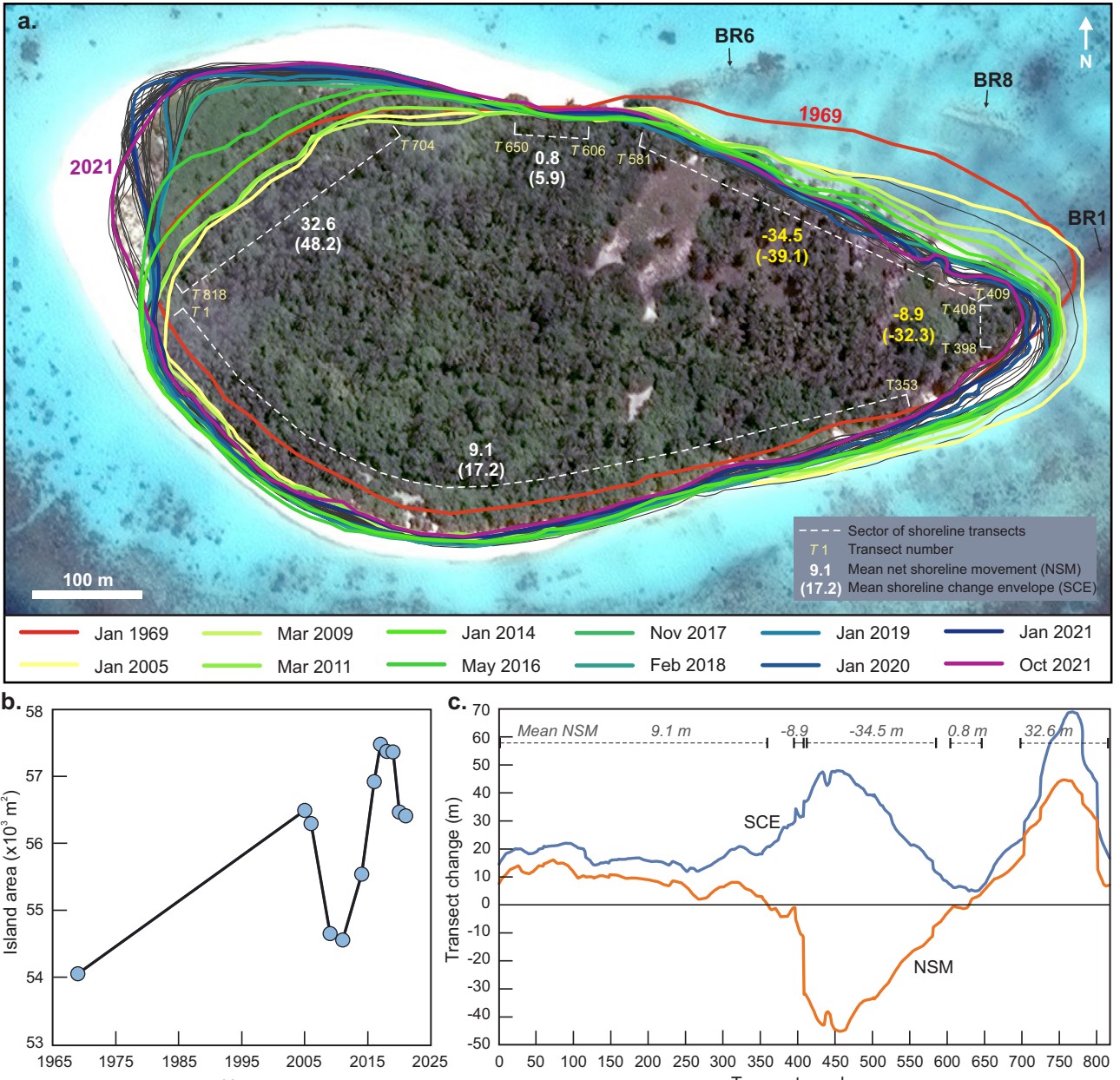

**Fig. 4 | Summary changes in the Kandahalagalaa shoreline 1969–2021.**
**a** Planform changes in the vegetated shoreline. Note summary values of net shoreline movement (NSM) and shoreline change envelope (SCE), in brackets, for specific sectors of the island (dashed white lines). Imagery © 2019 Maxar Technologies. **b** Summary changes in net island area 1969–2021. Data contained in Supplementary Table 4. **c** Results of DSAS shoreline change calculations for 818 shore perpendicular transects around the island. Source data[37].

Kandahalagalaa has been physically dynamic, altering its position, size, and shape, on the reef platform since its initial formation (Fig. 5). Of interest in this study is whether the magnitude of the recent change is unusual in the context of historical island dynamics and if so what are the possible drivers of such change.

Morphostratigraphic and radiometric evidence indicate that Kandahalagalaa formed across a shallow lagoon (faro) that had filled with sediment around 2400–2000 years ago, in accord with the faro infill model of island formation described elsewhere in the Maldives archipelago[26,30,35]. Island formation above the level of the lagoon/moat surface at Kandahalagalaa began after 1800 yBP. The primary phase of island construction, in which the island emerged above MSL, spanned the period 1500 to 1000 yBP, with subsequent expansion of its footprint occurring through to ~300–500 yBP (Figs. 2, 5).

The magnitude of post-formation island change is apparent in the spatial pattern and depositional context of radiometric ages. Data from core 13 on the central northern shoreline, and specifically the temporal coherence of ages from 2527 cal yBP at the base to 1344 cal yBP near the surface, indicates this northern sector of the island has remained stable since initial deposition. In contrast, a striking feature of the radiometric dates from the majority of other cores is the presence of age inversions (Fig. 2), indicating significant remobilisation and redeposition of island materials throughout the 1000 years since initial island formation. For example, in core 9 a radiometric age of 332 yBP lies ~3 m below the island surface with ages increasing up the core to 1133 yBP near the surface (Fig. 2a). The occurrence of the youngest age at depth signifies deposition of this portion of the island occurred after 332 yBP with older reworked sediment, or sediment from the

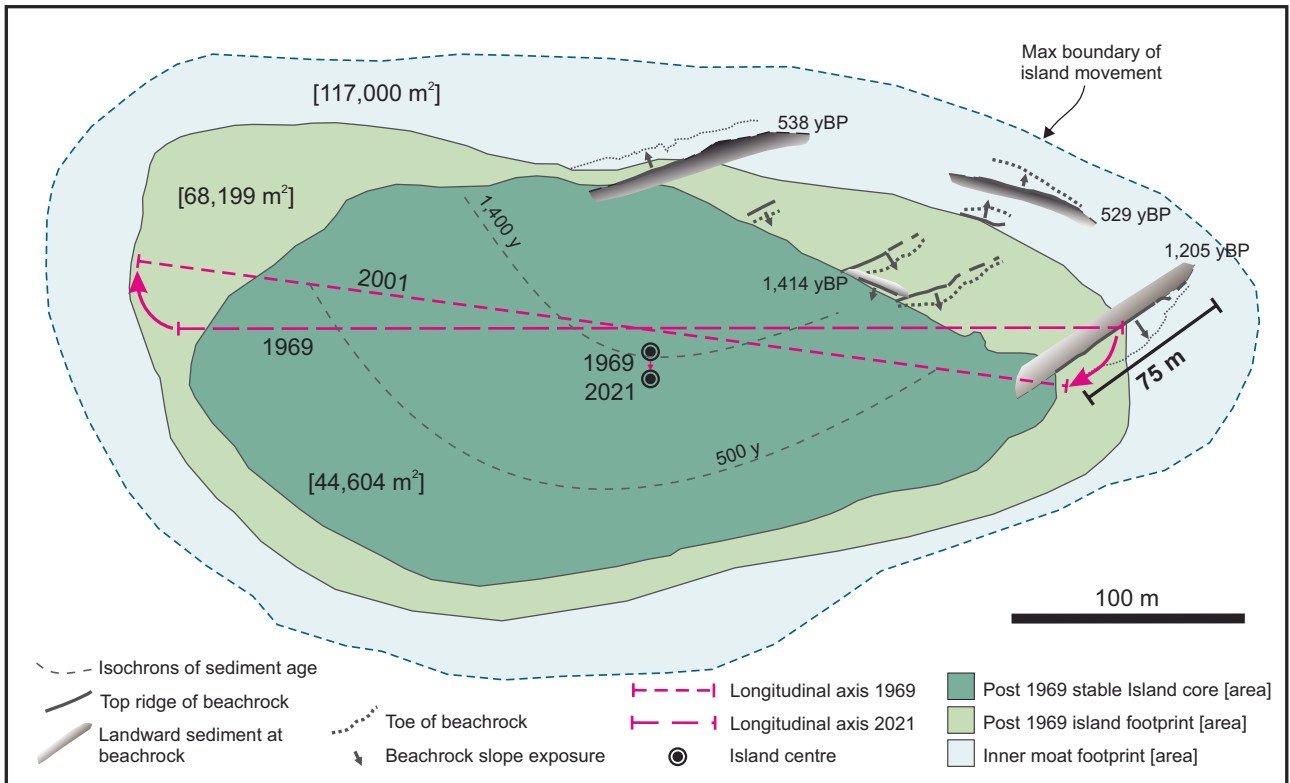

**Fig. 5 | Summary island change indicators at Kandahalagalaa, Huvadhoo atoll, Maldives, based on multi-temporal indicators and showing boundaries of large-scale movement of the island since formation ~1500 years ago.** Isochrons of equivalent age denoting youngest radiometric ages in cores, showing southward expansion of the island.

surrounding moat, deposited over this sample. The prevalence of active reworking of island sediments has effectively eliminated physical evidence of the initial extent of the island. However, isochrons on youngest ages indicate initial island formation in the north (core 13) around 1400 years ago, with subsequent expansion, through remobilisation and redeposition of sediments, across the southern moat surface by up to 150 m as recent as 500–332 years ago (Fig. 2; Fig. 5).

The presence, exposure and orientation of beachrock outcrops also provide clear evidence of positional mobility of the island by ±150 m (west to east) and a further 60 m offshore, over centennial timescales. The oldest ages on beachrock date to approximately 1440 yBP, consistent with the timing that the island emerged above MSL. However, the southward sloping beachface of this shore-parallel unit implies the island material must have been situated to the north of this outcrop with no island sediments located to the south when this outcrop was formed. There are a set of subsequent beachrocks, oriented oblique to the shoreline, that indicates episodic, and temporal, movement of the shoreline by 150 m to the east (beachrocks 5, 4, 2) within a 200-year period, terminating in the robust unit at the eastern end of the island dated at 1100 yBP (beachrock 1, Fig. 3a–c). This suggests that as early as ~1000 years ago the island had expanded to the east and substantive island materials were located to the northwest of this SE beachface sloping unit (Fig. 5).

There is a 600-year temporal hiatus to the final set of beachrock units which date to around 530 years ago and are located to the west (beachrock 6) and offshore (beachrocks 7 and 8; Fig. 3a, d–g, Fig. 5). The discrete ages of these units suggest they formed after a later phase of sediment generation and island mobility with the island sediments being located to the south of the detached beachrock units 7 and 8 (60 m from the contemporary shoreline) and to the east of beachrock number 6.

Collectively, analysis of radiometrically constrained island stratigraphy combined with geological markers of past shoreline positions indicates the island shoreline has historically fluctuated in position by up to ±200 m. Such magnitude of island change is greater than the shoreline changes observed over the past half-century based on satellite image analysis (Fig. 4) which establishes: accretion along the southern shoreline (-10 m); substantial erosion on the eastern end (−32 m) and northeast shoreline (max of −45.2 m, mean of −39 m); and, similarly substantive accretion on the northwest-western end of the island (max of 44.7 m and mean of 42.5 m; Fig. 4c). Values of the shoreline change envelope (SCE), that define the range of change at an individual transect, reach up to 70 m. However, these large short-term excursion distances are comparable in magnitude to the changes defined by the beachrock of ±150 m at century timescales.

Significantly, there has been a negligible movement of the shoreline on the central northern shoreline, proximal to beachrock 6 (Figs. 3a, 4c). The large gross changes in shoreline on either side of this location suggests there has been a substantial flux of sediment past this point and it may act as the fulcrum for island rotation. This location is also close to island core 13, which did not exhibit any radiometric age inversions and supports the hypothesis that this part of the island may have remained stable since island formation with subsequent expansion south (Fig. 5).

Aggregated at the island scale the net effect of the spatial differences in localised erosion and accretion of the shoreline is a southwesterly migration of the island and rotation toward the northwest (Figs. 4 and 5). It is particularly noteworthy that despite the significant shoreline reorganisation over the past half-century, Kandahalagalaa increased in area by 4.4% (2360 m²) highlighting that active island mobilisation during a period of sea-level rise is not necessarily synonymous with island erosion.

The island-scale envelope of change, defined by the vegetated footprint of the island, reveals two additional properties of island behaviour that are noteworthy in the context of island dynamics. First, the portion of the vegetated island that has remained stable over the past half-century (44,604 m$^2$) represents 80% of the initial island area, and this indicates that 20% of the 1969 island area has been transformed over the past 52 years. Second, the area of the reef platform occupied by the island over the past five decades has been 150% greater than the stable island core and reinforces the inherent mobility of the island over a relatively short timeframe and is consistent with historical interpretations of change. These findings suggest that in mobile island settings such as Kandahalagalaa, defining the differences between the stable and mobile island sectors can be instructive to inform future planning and construct adaptation strategies.

The large excursion in island footprint over recent decades, combined with beachrock outcrop positions, suggests that over longer timescales the island has migrated over a larger area of the reef surface than documented in short-term records. Indeed, the inner moat provides a geomorphic footprint that encapsulates the likely boundary of island migration over millennial timescales (Fig. 5). Unlike the sand and coral patch zone seaward of the moat (Fig. 2), the inner moat is characterised by a sand surface devoid of living coral. Previous studies have established such zones emerge due to periodic occupation of the moat surface by the island and beach, which precludes active coral colonisation and growth[18,26]. Ultimately, the inner moat can be considered a paleo-boundary that defines the long-term spatial dynamism of an island. On Kandahalagalaa the inner moat surface area of paleo-island movement is 117,000 m$^2$ (>200% larger than the current island area). Notably, the beachrock outcrops and southern shoreline are situated within 20 m of this boundary, and the recent footprint of island change indicates the shoreline is less than 100 m from the moat boundary at any point (Fig. 5).

In addition to the large-scale dynamism of Kandahalagalaa across decadal to millennial timescales, geomorphic evidence also provides insights into the causative mechanisms of observed island change. Previous studies in the Maldives archipelago identified seasonal monsoon variations as an important control on the local nearshore process signature (e.g., wave climates) which drive annual variations in beach position around islands, though these changes had little impact on the vegetated shorelines[18,38,39]. However, significant variations in the strength of the monsoons, that alter localised coastal processes are likely to impact the seasonal oscillation of beach position and differentially expose the vegetated shoreline to remobilisation. The Indian Ocean Dipole (IOD) is a regional coupled climate-ocean phenomenon that modulates changes in water level, winds and waves over inter-annual timescales[40–42] which may trigger island change. However, despite anecdotal evidence, the precise impacts of the IOD on nearshore processes along the archipelago and its influence on patterns of island erosion and accretion have yet to be resolved. In short, the enduring impact of such climatic anomalies in triggering periods of island instability is unknown. While unrelated to IOD processes, extreme events, such as tsunamis, have been shown to generate island instability for periods up to a decade[43,44]. At longer timescales, sea-level change is known to force significant changes in coastal systems[45,46], and has been implicated as a likely major cause of shoreline change in island settings. Geomorphic evidence from Kandahalagalaa suggests sea-level variability over the past millennia[34] and its associated influence on wave processes, may have played a significant role in observed island dynamics. Beachrock outcrops 7 and 8, which retain their beach slope and are contiguous units that have maintained their structural integrity (Fig. 3a, g), are ~0.5 m below MSL and well below the intertidal zone necessary for cementation of the beachface sediments. This elevation suggests these outcrops formed at a sea level lower than the present. Recent reconstruction of sea-level variability from Huvadhoo atoll has revealed an oscillating pattern of sea-level change over the

past two millennia[34]. Indeed, two periods of lower sea level (-0.8–0.9 m below present) coincided with the Late Antique Little Ice age and the more recent Little Ice Age (400–200 years ago). The age of sediments in beachrocks 7 and 8 are coincident with a fall in sea level between 500 and 400 years ago and this timing aligns well with the subsequent formation of the lower elevation beachrock, recording the lateral extent of shoreline progradation at that time (Fig. 5). The subsequent increase in sea level from this lowstand over the past 200 years is likely to have remobilised the unconsolidated beach materials and resulted in the displacement of the shoreline toward the south. Furthermore, ongoing sea-level rise at a rate of 3.46 ± 0.25 mm.y$^{-1}$ throughout the archipelago[47] may continue to drive mobility of the island at a pace similar to that of the past half-century. The migration of island shorelines away from exposed reef crests in response to increased sea level is consistent with a number of other studies of island change[12,16,17].

Our results highlight the value of adopting a multi-temporal methodological approach to gain a more complete understanding of the physical dynamism and temporal trajectory of reef islands. Significantly, this approach highlights that the magnitude of recent physical changes observed at Kandahalagalaa has not been unprecedented over the long term. Indeed, the range of paleo-dynamic evidence documented here shows large-scale changes in island dimensions, shape, elevation and beach levels as well as positional changes of ±150–200 m since island formation about 1400 years ago. Our findings are applicable to islands in similar depositional settings and indicate that recent trends of change observed globally on reef islands (Fig. 1a), are not likely to have emerged only in the past half-century but reflect an ongoing adjustment to environmental boundary conditions including sea-level change. Such centennial-scale changes in island morphology provide a historical setting for the many shorter-term studies that have emphasised shoreline shifts (erosion and accretion) over the last few decades primarily based on sequential satellite imagery. Whilst this short period spans the personal experience of atoll residents who have witnessed these changes, and who may have undertaken local adaptations to reduce impacts[48], we have shown that the magnitude of recent island movements has been dwarfed by those that have occurred over the previous centuries. Significantly, while there has been recent enthusiasm for creating nature-based solutions, that work with environmental processes, to support climate change adaptation in reef island settings[49], many proposed solutions still rely on a temporally short-sighted understanding of system dynamics, rely on vegetation solutions or technological interventions, and ignore landform dynamics. Such considerations need to be based on more than observations over recent decades and should be contextualised where possible, against the longer-term dynamics of island landforms. However, currently, the number of studies with sufficiently mature datasets to perform comparable, or indeed comparative analysis, is lacking. Our results highlight the importance of generating such datasets in understanding the natural dynamics of islands over planning and longer time- scales, the timescales of relevance to human security, and how communities can incorporate current and future land changes into their adaptation trajectories. Results from the multi-temporal analysis of island dynamics, adopted in this study, provide a robust empirical basis of landform change to support informed decision-making and the development of adaptation pathways.

## Methods
### Field Site
This study examines the temporal dynamics of Kandahalagalaa, a lagoonal reef platform island located in Huvadhoo atoll in the south of the Maldives archipelago, central Indian Ocean (Supplementary Fig. 1). The ocean wave climate of southern Huvadhoo is influenced by local wave processes generated by seasonal monsoon winds, periodic storms and long period swell waves generated by large low-pressure systems that originate in the Southern Ocean[47,50]. A seasonal

distinction can be made in the wave climate, with lower swell and northeast winds at the study site persisting during the northeast monsoon (between December and February), and a mixture of southwest wind-driven waves and larger long-period swell during the southwest monsoon (between April and September). Significant wave height offshore of Huvadhoo is typically just over 1 m during the northeast monsoon and ~1.95 m during the southwest monsoon. However, there can be significant variability in Hs values about the mean with maximum monthly Hs values ranging from ~2 m in the northeast to ~3.8 m in the southwest monsoons, respectively. The island is located less than 5 km inside the atoll rim, which has deep passages connecting the lagoon and ocean to the E, SSE and West (Supplementary Fig. 1). Consequently, the study island experiences a lower wave energy regime which is an ensemble of residual ocean swell that propagates through the deep passes and internally generated lagoon wind waves. Throughout the archipelago, the oscillating monsoon seasons impart distinct wave and current signatures around island shorelines that drive beach dynamics characterised by westward movement under the influence of the northeast monsoon and eastward movement under southwest monsoon conditions[18,39]. The field site is located close to the equator and consequently, is not affected by frequent extreme wave events. However, larger storms do infrequently occur[47], which can flood nearby islands.

Analysis of sea-level records since 1987 from Gan in the southern Maldives[47] indicates that the sea level in the southern archipelago has increased at a mean rate of $3.46 \pm 0.25$ mm.y$^{-1}$. Interannual oscillations in mean sea level (MSL), influenced by climate phenomena such as ENSO and the Indian Ocean Dipole (IOD) are also present in tide gauge measurements with an amplitude in the order of 0.2 m. The atoll is subject to a semi-diurnal tidal regime with a spring tide range of 0.96 m with a pronounced diurnal inequality[47].

The triangular-shaped Kandahalagalaa reef platform is 26.9 ha in area (Supplementary Figs. 1c and 2). Kandahalagalaa island formed in the late Holocene[33] during a period when the sea level was falling from a highstand 0.5 m above present and has persisted on its reef platform during a period in which there has been minor oscillations in sea level of up to 0.9 m over the past two millennia[34].

Topographic surveys show the island is positioned on an inner and shallow basin on the central platform, and that there are several distinct eco-geomorphic zones, that are significant in evaluating the foundations for island formation and that delimit the spatial mobility of the island (Supplementary Fig. 2). The central platform, and island, is encircled at the seaward extent by a higher elevation outer coral zone that consists of a dense network of dead and living coral framework (Supplementary Fig. 2). Covering an area of 148,826 m$^2$ (38.1% of platform) this zone varies in elevation from −0.63 to −0.86 m MSL, with isolated corals up to −0.43 m MSL. Ranging in width from 50 m (north) to ~100 m the outer coral zone terminates abruptly at its lagoonward extent with a marked transition onto a lower elevation sanded reef pavement surface. Ranging in elevation from −1.45 to −1.73 m MSL this zone has conspicuous and isolated coral patches that reach elevations of ~ −0.63 m, like the outer coral zone, and which decrease in frequency toward the island (Supplementary Fig. 2). Notably this zone is absent along the southern shoreline. Immediately surrounding the island shoreline and ranging in width from 25 m (south) to more than 100 m (west and east) is a sanded reef surface (the inner moat) that is devoid of coral growth forms and ranges from −1.0 m MSL (east) to −1.55 m (west). The lower elevation inner sandy moat covers an area, including the island, of 117,024 m$^2$ (30.4% of platform surface).

The vegetated island and beach is 81,360 m$^2$ in area (20.8% of platform area) with surveys along the north–south axis showing the island surface is near-planar, characterised by five low amplitude ridges ranging in elevation from 1.39 m MSL on the northern shoreline to 1.56 m MSL on the southern shoreline (Fig. 2a). The island has a more asymmetric topography along its west–east axis (Fig. 2b), increasing in

elevation from 1.06 m MSL on the western island ridge, onto a central surface that is 1.43 m MSL and terminating on the eastern ridge which has a maximum elevation of 1.9 m MSL, the highest surveyed point on the island, which aligns with the deep pass through the atoll rim (Supplementary Fig. 1c).

## Reconstructing island change

This study examines the development and spatial dynamics of the island at three distinct temporal scales.

**Millennial-scale island formation.** The evolutionary dynamics of the island in the late Holocene are examined based on the island morphology and sedimentary structure, which is temporally constrained by radiometric dating. The topography of Kandahalagalaa reef platform from reef edge to island surface was surveyed along four transects (Supplementary Fig. 2a). The island morphology was characterised based on six survey transects (Supplementary Fig. 2a). All surveys were conducted using a laser level, and surveys were reduced to mean sea level using sea-level records at Gan (00°41S, 73°09E) accessed through the University of Hawaii Sea Level Center.

Subsurface stratigraphy was reconstructed by analysing the skeletal composition and textural properties of 154 sediment samples from 20 cores extracted across the island (Supplementary Fig. 2a) using percussion coring and augering techniques[33]. Stratigraphy of the island was temporally constrained with 27 radiometric ages using the green calcareous algae *Halimeda* (23 samples), bulk sand (3 samples), and a coral stick from the base of core 12[33]. An additional three percussion cores were retrieved from the northern reef edge and nearshore moat on the northern and southern shorelines (Supplementary Fig. 2a) to place the island chronology in the context of the surrounding moat and reef surface. Textural analysis of 26 samples from these cores was undertaken using techniques consistent with previous studies[33]. An additional five radiometric dates were obtained from these cores to constrain the timing of reef accretion and sedimentation in the lagoonal platform. Radiocarbon dates were obtained from the Radiocarbon Dating Laboratory, University of Waikato (Wk), New Zealand, and Direct-AMS, USA. Ages were calibrated using OxCal version 4.4[51] with Marine 20 curve[52] and Delta-R (−46, 51) as the best estimate for the central Indian Ocean. All radiometric ages are presented in Supplementary Table 2.

**Centennial to millennial-scale island dynamics.** At the centennial to millennial timescale, a sequence of conspicuous beachrock outcrops was examined to reconstruct paleo-shoreline positions. Beachrock forms through the lithification of beach sediments by calcium carbonate cements in the intertidal zone and provides an indicator of past shoreline position, orientation and slope[36]. At Kandahalagalaa, a set of lithified beachrock outcrops were identified on the northern and eastern sides of the island, a number of which are detached from the existing shoreline (Supplementary Fig. 2a). The orientation of each outcrop relative to the north was identified from satellite imagery, while topographic surveys across each outcrop determined its elevation, slope and beachface orientation. Surveys were also reduced to mean sea level (MSL). Small cores were retrieved from each outcrop using a handheld drill and examined to assess the fabric of the beachrock. Radiometric ages were determined on 11 beachrock samples. Specifically, dates were determined on the bulk sand matrix of the beachrock and thus only provide an indication of the earliest possible time of formation associated with the death of organisms. Radiometric ages are presented in Supplementary Table 2.

Visual inspection of the fabric of the beachrock shows it is comprised of medium to coarse sands, that are texturally and compositionally indistinguishable from the island and beach materials. In general, the outcrops all had significant sections that were coherent and retained beach slope structure with visible

inclined bedding planes (Fig. 3). The most seaward sections of several beachrock outcrops exhibited some cracking and slabbing, had lost their structural integrity and were slumped on the sandy moat surface. These slumped parts of the beachrock outcrops have been exposed to weathering processes for a prolonged period, whereas landward portions were likely buried underneath island sediments and have subsequently been exhumed and exposed by more recent reshaping of the shoreline.

**Multi-decadal island change.** To examine more recent multi-decadal scale changes in the Kandahalagalaa shoreline, we compare shoreline positions reconstructed from a historic aerial photograph taken in 1969 with commercial multispectral satellite images captured by Airbus and Maxar providers. Images sourced from Airbus were captured by the Pléiades satellites (50 cm resolution). Images sourced from Maxar were captured by the QuickBird-2 satellite (Jan. 2005, 60 cm resolution), GeoEye-1 (Mar. 2009, 40 cm resolution) and WorldView-2 satellite (all other images, 50 cm resolution). In total, 12 images were analysed with the analysis window spanning 52 years (1969, 2005, 2006, 2009, 2011, 2014, 2016–2021). Multispectral satellite imagery was obtained as true colour images which had already been pan-sharpened prior to delivery. Pan-sharpening is a process through which the coarser resolution multispectral imagery is fused with higher-resolution panchromatic imagery captured simultaneously to sharpen the multispectral images. The pan-sharpened images have a spatial resolution between 40 and 60 cm. All satellite imagery was provided georeferenced; however, further georeferencing was required to improve alignment between images. A Maxar image captured in 2019 provided the source of ground control points for georeferencing imagery and the 1969 aerial photograph. The 2019 image was selected as there was very little glare on the water making identification of subtidal features often used for ground control points easier. Given the paucity of stable anthropogenic features on most islands, natural features such as lithified beachrock were used as ground control points following similar studies in other atoll settings.

Replicating similar studies of reef island change, the edge of vegetation is used as a proxy for the island shoreline[12,15,53]. The edge of vegetation is readily identifiable in all imagery, represents the vegetated core of the island, and filters short-term noise associated with the interpretation of more dynamic beach shorelines.

Three sources of uncertainty were considered when calculating the positional uncertainty in the edge of vegetation, being: georeferencing, pixel and digitising errors[53]. Georeferencing error was derived from the root-mean-square error from the georeferencing of each image. The spatial resolution of scanned aerial photographs and satellite imagery represents the pixel error. We adopted a digitising error of 1.6 m, which was derived from repeated digitising of vegetation lines on reef island shorelines[54]. Total shoreline error (Te) was calculated as the root sum of all shoreline positional errors and ranged between 1.64 and 2.23 m for the shorelines interpreted from satellite imagery and 6.07 m for the shoreline interpreted from the 1969 aerial photograph.

Shoreline change analysis was undertaken using the Digital Shoreline Analysis System (DSAS), an extension within the GIS software package ArcMap[55]. DSAS analyses change by recording the intersection of transects cast perpendicular to a user-generated baseline and the shorelines. In this study, transects were cast every 1 m along the baseline with a total of 818 transects analysed around Kandahalagalaa. A number of physical change statistics were then automatically calculated using the position of the intersection of shorelines and transects. Three measures of island change were examined. First, the shoreline change envelope (SCE) captures the total excursion of the shoreline at each transect. Second, net shoreline

movement (NSM), calculates the net shoreline movement between the initial (1969) and final (2021) shoreline positions. Third, the annualised rate of change between the initial and final shorelines, known as the end point rate (EPR) was calculated. Given the multi-decadal timeframe of the dataset, the EPR is expressed as the decadal rate of change (m per decade). A confidence interval of 2σ (95.5%) was applied when calculating shoreline change rates. Transects with statistically significant rates of change are considered erosional (negative EPR) or accretionary (positive EPR); the remaining transects are classified as exhibiting no detectable change.

## Data availability
All data generated or analysed during this study are included in the published article, its supplementary information file, and a public data repository. The data in Figs. 1a and 4c are available in the Zenodo Research Data Repository under accession code https://doi.org/10.5281/zenodo.7471194.

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

## Acknowledgements

We acknowledge LaMer Group and the Small Island Research Centre, Fares-Maathodaa, Huvadhoo atoll for logistical support and the Government of the Maldives for research permission under the Ministry of Fisheries and Agriculture permit number 30-D/INDIV/2018/28.

## Author contributions

P.K. conceived the project; P.K., C.L., S.O., E.R., T.T., E.B., M.D., W.S., A.M., A.V.-C. and R.M. undertook fieldwork; P.K., C.L., E.R., T.T. and M.F. undertook data analysis; P.K. led manuscript development and interpretation and all authors contributed to manuscript revision.

## Competing interests

The authors declare no competing interests.
