## [Peer Review File · Nature Communications]

REVIEWER COMMENTS

Reviewer #1 (Remarks to the Author):

Overview

Kench et al (2022) sets out to resolve whether recent documented changes in the size, shape and location of coral reef islands are unprecedented compared with the pre-industrial era. They use a range of different techniques to do so such as radiometric dating and remote sensing. In my opinion, this is a great and really interesting paper, providing novel results on the subject matter. I have very few comments (provided below) that the authors might want to consider. I believe these changes would improve the manuscript, but the authors are in the best position to decide how to present their work.

General comments

As currently written, the manuscript is detailed and specific, reading like a field report. This is of course not a bad thing. However, as the manuscript has been submitted to Nature Communications, the authors might want to consider modifying the manuscript so it is suitable for a broader audience.

The results and discussion section are both quite long, and there is some repetition of the results in the discussion. I think this section would be best reserved for the implications of the work.

In the 4th paragraph of the introduction, several results are described, e.g.:
"Fourth, contraction of islands through erosion has been observed in <34% of islands"
Please provide references for such values if they are from other papers.

In Fig. 1, island area is described in hectares. Would the use SI-unit be more appropriate here?

Minor comments

Beach slope is described in terms of percentage. Is this intended and preferred over degrees?

A general proofread is needed to pick up some minor issues e.g. instances of hyphens (-) used where an en dash is needed (–).

Overall, congratulations on putting together a really nice piece of work!

Signed

Dr Lewis A. Jones
Centro de Investigación Mariña
Universidade de Vigo

Reviewer #2 (Remarks to the Author):

This is undoubtedly a topic of considerable scientific and global political interest and this is a team with a great track record in this field for rigorous field-based research. It asks an important question – the assumption of island stability prior to more recent SLR acceleration – that does need proper testing. And it makes a good point about the static base of modelling studies versus the dynamics that come out of remote sensing and field-based monitoring and the shortcomings of a purely radiocarbon-based analysis. The summary argument (lines 330-351) is well made.

But for all that, the manuscript is rather unsatisfactory in its present form. It seems a shame to wrap (and it happens very suddenly at line 107...) some very interesting questions around just another, albeit very detailed, study of environmental history on a

single atoll island. Bluntly, it is not as if this hasn't been done before and indeed in the Maldives before. Also, that detailed study is really about island vertical stratigraphy and the time development of a reef island on an atoll margin. That does not sit easily within a broader argument about changing island area and position largely restricted to the last 1,500 years from the beachrock evidence. The authors do their best, through discussions of the location of cores with age inversions, but all they can really say is that the island has been dynamic over time, surely not much of a surprise to many readers. Better, perhaps, to accept island formation at ca. 1,500 yrs and then concentrate on the beachrock and satellite-derived shoreline change evidence and arguments.

Why was Huvadhoos chosen? And why Kandahalagalea on Huvadhoos? More broadly, there are of course several locations where this kind of detailed analysis has been undertaken in the Indo-Pacific reef province. Would it not have been more powerful to look at this argument on multi-temporal scaling across a number of sites (ideally plotting in different positions on Figure 1a) than concentrating, in great detail, on a single site? It may be that the paper has to be structured in this way but it would be useful to have information on tidal range, wave climate (and its seasonal variation?), water level variability (IODP etc. – doesn't appear until lines 301-308), historical and current rates of SLR (finally at 326-327) before diving into the detail of Kandahalagalea island stratigraphy.

Structurally, the journal requires a certain organization. This perhaps does not help the argument in this instance but the structure is what it is. However, even within the main text the argument might be organised better. The context of concerns over atoll futures (including the simplistic projections) – then what the historical (last ca. 1,000 years) and current / recent (50 years) record shows – then is the recent unusual? The paper could be more focussed on these questions and the evidence base relating to them. In places, the general argument (lines 51-64) is not entirely objective. What would constitute stability? +/- 10% change of area? What % would fall into this category? The paper is well written (if a bit dense when it comes to the stratigraphic and morphological detail) with an appropriate level of referencing but it has a habit of telling the reader that something is important without actually spelling out what that importance is; some examples are picked up in the specific comments below (e.g. lines 31, 34 and 43). This is particularly the case when the reader is told repeatedly that this study has important implications for the planning of adaptation strategies or informing 'nature-based solutions' without ever stating what those strategies and solutions might be. It is not until lines 287-288 that it is made clear that there is value in planning terms in identifying the stable and mobile sections of an island for adaptation planning purposes. It never explains how the undefined 'nature-based solutions' fit into the board argument on multi-temporal island dynamics.

Specific points

16-17: Surely the concerns are really about future change rather than recent historical change? When you get to the start of the Introduction the argument is slightly different and more forward-looking.

20-23: the argument is made in terms of magnitude. But if you convert this to a rate then the picture looks quite different – a rate of change of 0.8 m a⁻¹ over the last 50 years v. a rate of 0.1 m a⁻¹ over 1,500 years. Or is rate a function of the smoothing effect of a longer timescale.

31: 'trigger substantial changes in the physical structure' – what exactly is expected?

34: 'a more viable set of future outcomes' – what exactly might be the nature of this set?

43: what are the 'physical process mechanisms'?

47-48: 'Such information is fundamental to inform robust adaptation planning'. How is it fundamental?

59: what % is 'rare'?

67-70: there is some repletion of argument here from 44-47

72-80: Figure 1a is difficult to read as it conflates x4 image analysis periods. Would it be better to have 4 panels (with Huvadhoos on the appropriate panel) , one for each of the image periods? On Figure 1b, it would be more helpful to have the different reef areas

rather than the individual atolls. Where does Huvadhoon best fit into this plot?

85-88: the text timeframe of island accumulation, with its three modes of discrete formation, continual process or discrete phases is not well linked to Figure 1b. How do we see these different modes in the Figure? Can examples be given?

89-90: the text on community experience and adaptation seems speculative. How does the arrival of particular peoples in particular locations relate to particular modes of island development and change? Can the anthropologists supply examples? And, specifically, how is this relevant to Kandahalagalea?

104-106: the text anticipates the results here

138-142: it seems odd that there is no locational map for these two transects and that it has been hived off to Supplementary Figure 2A

195: so a 4% difference from 1969 area. So to all intents and purposes = stable?

205-207: what are the boundaries between erosion / stable / accretion – what are the defining rates?

260: so are we seeing the 'discrete phases' model of island accumulation? Needs stronger relation to text around lines 85-88

343: what are these 'nature-based solutions'?

Reviewer #3 (Remarks to the Author):

The paper provides a tidy and detailed study of the evolution of a Maldivian low reef island. It uses

- the work is of significance to the geo-, bio- and social sciences. It is an important contribution to inform on island persistence under global change and so also has political relevance (esp. for small-island states).
- the work supports the conclusions and claims. No further analysis is needed.
- there are no flaws in the data analysis, interpretation and conclusions.
- the methodology is sound
- there is enough detail provided in the methods for the work to be reproduced.

The text is concise, short yet long enough to provide all needed information. It is very polished and I have no issues here at all. I can clearly follow the arguments.

Graphics are high-quality and suitable for the publication. They should not be shortened.

I found only 2 spelling mistakes:

line 137: change "millenia" to "millenium". I think the authors want to say "in the last 1000 years", so the singular, millenium, needs to be used. Millenia is the plural and would mean "in the last several thousand years".

line 170: change to "...two beachrock outcrops", deleting the plural s at the end of beachrocks.

Some more information would be welcome in the methods:

- radiometric dating: please provide more detail on the actual method, if the allowed space permits.
- satellite imagery: please provide some more information on what type of imagery (Maxar and Airbus are just the providers), i.e. from which sensor (GeoEye? Sentinel? what pixel resolution?). It would also be interesting to know how that 1969 aerial imagery was geo-referenced to the new images (which come geo-referenced). Using the same method as described for the satellite images?
- Shoreline calculation. I can follow what was done in the software. But the shoreline changes with tidal state and images are acquired at different tidal states. How was that corrected?

Response to Review Comments

Reviewer #1.

Overview

Kench et al (2022) sets out to resolve whether recent documented changes in the size, shape and location of coral reef islands are unprecedented compared with the pre-industrial era. They use a range of different techniques to do so such as radiometric dating and remote sensing. In my opinion, this is a great and really interesting paper, providing novel results on the subject matter. I have very few comments (provided below) that the authors might want to consider. I believe these changes would improve the manuscript, but the authors are in the best position to decide how to present their work.

We thank the Reviewer for their supportive comment and appreciate they saw the novelty in the research.

General comments

As currently written, the manuscript is detailed and specific, reading like a field report. This is of course not a bad thing. However, as the manuscript has been submitted to Nature Communications, the authors might want to consider modifying the manuscript so it is suitable for a broader audience.

We have been mindful of this comment in our revisions. In particular, we have been able to summarise the historical island change section more succinctly and provide an improved transition from the introduction to the results section.

The results and discussion section are both quite long, and there is some repetition of the results in the discussion. I think this section would be best reserved for the implications of the work.

We have been more economical with presentation of both results and discussion with the aim of reducing overlap. However, we argue that is necessary to highlight aspects of the results in developing the synthesis and to interpret and compare island change at different timescales.

In the 4th paragraph of the introduction, several results are described, e.g.:

“Fourth, contraction of islands through erosion has been observed in <34% of islands”

Please provide references for such values if they are from other papers.

These are interpretations from the dataset presented in the Fig. 1A. We have referenced this figure after the mention of 34% of islands.

In Fig. 1, island area is described in hectares. Would the use SI-unit be more appropriate here?

We initially chose to use hectares, as this is the unit used in the vast majority of studies of island area change over the past decade. However, we have amended the scale to square kilometres.

Minor comments

Beach slope is described in terms of percentage. Is this intended and preferred over degrees?

Beach slope can be described either by use of degrees or percentage. In this study the orientation of the beachrock outcrops relative to the shoreline is described using degrees. Consequently, to avoid confusion, we adopt percentage values to distinguish orientation of the beachrock versus the slope of the beachrock.

A general proofread is needed to pick up some minor issues e.g. instances of hyphens (-) used where an en dash is needed (–). Thank you. *We have gone through the manuscript and sought to correct these instances.*

Reviewer #2 (Remarks to the Author):

This is undoubtedly a topic of considerable scientific and global political interest and this is a team with a great track record in this field for rigorous field-based research. It asks an important question – the assumption of island stability prior to more recent SLR acceleration – that does need proper testing. And it makes a good point about the static base of modelling studies versus the dynamics that come out of remote sensing and field-based monitoring and the shortcomings of a purely radiocarbon-based analysis. The summary argument (lines 330-351) is well made.

We thank the Reviewer for these positive comments. We are pleased the Reviewer agrees with the main point of the argument related to understanding whether recent changes are unprecedented over the longer period of island existence.

But for all that, the manuscript is rather unsatisfactory in its present form. It seems a shame to wrap (and it happens very suddenly at line 107...) some very interesting questions around just another, albeit very detailed, study of environmental history on a single atoll island. Bluntly, it is not as if this hasn't been done before and indeed in the Maldives before. Also, that detailed study is really about island vertical stratigraphy and the time development of a reef island on an atoll margin. That does not sit easily within a broader argument about changing island area and position largely restricted to the last 1,500 years from the beachrock evidence. The authors do their best, through discussions of the location of cores with age inversions, but all they can really say is that the island has been dynamic over time, surely not much of a surprise to many readers. Better, perhaps, to accept island formation at ca. 1,500 yrs and then concentrate on the beachrock and satellite-derived shoreline change evidence and arguments.

Thank you for this comment. We acknowledge that other island evolution stories have been published. However, we do not believe this comment is a fair reflection of the data presented. We believe the dataset presented differs from previous analyses in several ways. Indeed, such a study as presented in the manuscript has not been attempted to our knowledge. First, past studies have focussed primarily on the timing of the onset, and window of island formation often using a limited number of radiometric dates (commonly less than 10). The limited number of dated samples used in past studies, has prevented more detailed analysis of post-formation changes. Our study presents one of the most detailed radiometric dating datasets that crucially has multiple dates along cores. We use the collective dataset as a framework to explore further detail in the evolutionary evidence. Second past studies have not attempted to resolve post-depositional changes in islands based on analysis of the spatial pattern of radiometric ages and inferences on this pattern, largely due to inadequate sampling. Our evolution story does not simply note when the island formed, rather due to the comparatively rich radiometric dating data we are able to decipher, through the recognition of inversions and use of minimum ages in cores, significant expansion of the island footprint by up to 150 m southward across the reef platform with significant expansion as late as 500 years ago as summarised in the isochrons in Figure 5. Consequently, we believe the data does sit easily within the broader argument about changing island area and position and we consider we have further strengthened this point in the revised text. Third, no previous study has attempted to link the historical data with additional evidence of island change.

As suggested by the Reviewer we have condensed the treatment of island evolution prior to initiation of island formation, to enable a greater focus on post formation modifications. Furthermore, we have attempted to emphasise the value of the spatial pattern and vertical sequences of radiometric dates to decipher island change, to confirm the value of such analyses. Combined with our subsequent introduction of beachrock dates and aerial imagery presents a comprehensive overview of island dynamism.

Ultimately, the importance of this work is saying more than simply islands are dynamic over time and can place timeframes and magnitudes of change on such changes. We do believe that the magnitude of changes identified in the study will be a surprise to many readers, as we believe such analysis has previously not been attempted and as we argue, there is a disconnect between a contemporary focus on loss or accretion of the island edge and a historic focus on islands emergence and expansion. In part this disconnect may arise as historic traces of island change may

arguable be obscured by the movement of the island across old beach faces, revealing and covering relic structures over time.

Why was Huvadhooh chosen? And why Kandahalagalea on Huvadhooh? More broadly, there are of course several locations where this kind of detailed analysis has been undertaken in the Indo-Pacific reef province. Would it not have been more powerful to look at this argument on multi-temporal scaling across a number of sites (ideally plotting in different positions on Figure 1a) than concentrating, in great detail, on a single site? It may be that the paper has to be structured in this way but it would be useful to have information on tidal range, wave climate (and its seasonal variation?), water level variability (IODP etc. – doesn't appear until lines 301-308), historical and current rates of SLR (finally at 326-327) before diving into the detail of Kandahalagalea island stratigraphy.

There are several points/questions the Reviewer makes in this paragraph and we deal with each separately. First, the Reviewer asks why Kandahalagalea and whether additional sites from the Indo-Pacific could also have been included. We are not aware of any sites where the level of detail presented in our study are available. There are 1,100+ sites where there have been estimates of island change over recent decades using aerial imagery. However, in only a few dozen is there a high density of images to allow sub-decadal analysis of change. There are approximately two dozen studies that have presented island formation histories. However, in less than four of these studies have there been more than 20 radiometric dates able to constrain the window of island accumulation. In three of these remaining studies the dates are from substantially larger islands. Furthermore, in these studies there are seldom more than one or two radiometric ages from cores. Consequently, the ability to infer post-depositional reworking of the island materials at the island-scale, as we present in this study, is not available from other sites. Finally, we are unaware of any other studies which present detailed geomorphic and radiometric analysis of beachrock outcrops as paleo-shoreline markers, that are related to islands where island evolution has been attempted. As a consequence, for the above reasons, Kandahalagalea was chosen as it is a rare site where we have detailed data on evolution/stratigraphy, beachrock paleo-shorelines and decadal-scale planform change. We consider this an unparalleled dataset, that currently cannot be replicated from existing sites in the Indo-Pacific. Consequently, it is currently not possible to compare multiple sites, though this is clearly a future goal. We believe our analysis demonstrates the values of such multi-temporal analysis and should stimulate similar work through the Indo-Pacific.

We thank the Reviewer for raising this broader issue and have amended our manuscript to better contextualise why we chose this study site and why the data available sets it apart from other sites. We pay attention to the clarity of this message in the final paragraph of the introduction and the methods. We reinforce this message that based on the value of our analysis that comparable datasets should be generated to enable broader comparison across the Indo-Pacific. We highlight the necessity to re-examine and obtain higher density radiometric evidence from islands to assist in resolving multi-centennial island dynamics.

The Reviewer seeks additional data on tidal regime, wave climate and IODP. We have added such information to the methods/field site section. It is our understanding that such field description data is not a standard component of the main body of the text in the *Nature Communications* style. However, we are amenable to repositioning this in the main body of the text at the editor's discretion.

Structurally, the journal requires a certain organization. This perhaps does not help the argument in this instance but the structure is what it is. However, even within the main text the argument might be organised better. The context of concerns over atoll futures (including the simplistic projections) – then what the historical (last ca. 1,000 years) and current / recent (50 years) record shows – then is the recent unusual? The paper could be more focussed on these questions and the evidence base relating to them. In places, the general argument (lines 51-64) is not entirely objective. What would constitute stability? +/- 10% change of area? What % would fall into this category?

We thank the reviewer for this comment. Indeed, we had structured the manuscript in a similar fashion to that suggested beginning with the evidence for change at millennial, centennial and decadal scales that draws on the three primary datasets. We then draw these together in the discussion to address the last question. To ensure this structure is clear to the reader we have endeavoured to rework the text to provide an improved and more explicit framework for

this set of questions and signposted this more explicitly toward the end of the introduction and beginning of the results and discussion.

The paper is well written (if a bit dense when it comes to the stratigraphic and morphological detail) with an appropriate level of referencing but it has a habit of telling the reader that something is important without actually spelling out what that importance is; some examples are picked up in the specific comments below (e.g. lines 31, 34 and 43).

Thank you for identifying these cases. We have rectified these statements and provide specific examples to support the statements.

This is particularly the case when the reader is told repeatedly that this study has important implications for the planning of adaptation strategies or informing 'nature-based solutions' without ever stating what those strategies and solutions might be. It is not until lines 287-288 that it is made clear that there is value in planning terms in identifying the stable and mobile sections of an island for adaptation planning purposes. It never explains how the undefined 'nature-based solutions' fit into the broader argument on multi-temporal island dynamics.

We have amended the text to convey the value of data on island change to support management and adaptation strategies. Specifically, information of rates of island change, the magnitude of land change and trajectories of change (expanding in places or contracting in places) we consider fundamental to underpin any future land-use planning. This is not an article about nature-based solutions and we do not have the space in the manuscript to delve into this topical area. However, a number of recent articles have been published that seek to identify a raft of 'new' technical nature-based solutions. Seldom do such approaches consider the actual natural processes of island change. Our intent here is to highlight that the types of changes we can detect through our analysis can be of great value in planning.

Specific points

16-17: Surely the concerns are really about future change rather than recent historical change? When you get to the start of the Introduction the argument is slightly different and more forward-looking.

Thank you for raising this point. We have amended the sentence to also include the next century.

20-23: the argument is made in terms of magnitude. But if you convert this to a rate then the picture looks quite different – a rate of change of 0.8 m a⁻¹ over the last 50 years v. a rate of 0.1 m a⁻¹ over 1,500 years. Or is rate a function of the smoothing effect of a longer timescale.

We make the argument in terms of magnitude as rate implies a linear directional trend and we think is less able to convey fluctuations in position. Due to the nature of the long-term datasets we are unable to decipher such fluctuations, though the beachrock evidence clearly shows considerable variability in shoreline. We would not preclude historic rates of shoreline change occurring at equal or larger rates than the shorter-term observations indicate. However, such changes would be smoothed in a rate calculation as indicated by the Reviewer.

31: 'trigger substantial changes in the physical structure' – what exactly is expected?

We have amended the sentence to identify the key changes expected including, erosion, loss of freeboard and complete loss of islands.

34: 'a more viable set of future outcomes' – what exactly might be the nature of this set?

We have amended the sentence to indicate what these outcomes may include. Specifically, the sentence now reads: *'However, more recent studies using both field-based and modelling approaches indicate a broader suite of future outcomes for small islands and their communities, in which islands will continue to persist and remain available for habitation.'*

43: what are the 'physical process mechanisms'?

We have amended the sentence to explicitly identify mechanisms that drive island change including the alongshore redistribution of sediments, wave overtopping and overwash sedimentation.

47-48: 'Such information is fundamental to inform robust adaptation planning'. How is it fundamental?

We consider that adaptation strategies and land-use planning would benefit from sound knowledge of the magnitude and trajectory (style) of island change. We have amended the sentence to be more explicit in this regard.

59: what % is 'rare'?

We have now placed a value on this in the text. From the larger datasets of 1,100 islands, only three islands have been found to have been lost from atoll reefs (0.27%).

67-70: there is some repetition of argument here from 44-47

We recognise there is a minor repetition of the argument at this point. However, we believe this is a necessary revisiting of the point for continued development of the argument in the subsequent section.

72-80: Figure 1a is difficult to read as it conflates x4 image analysis periods. Would it be better to have 4 panels (with Huvadho on the appropriate panel), one for each of the image periods? On Figure 1b, it would be more helpful to have the different reef areas rather than the individual atolls. Where does Huvadho best fit into this plot?

We have considered this comment carefully. We specifically chose to display the different windows of time, as this differs from all previous meta-analyses that present change without temporal windows. We do not agree that the panel A is difficult to read and believe the intent of the panel is to show the complete dataset, and the variation. It is not the intent to be able identify specific islands and hence doubt the value of presenting four separate images of the same information. We also believe the colour differences presented are able to be detected and that generating four panels would be wasteful of space, where the journal has tight limits on the number figures able to be included in the main text.

Figure 1b does not have individual atolls. Rather it identifies specific islands on which island evolution has been studied. We do not believe presenting reef area in this panel would add any further context.

85-88: the text timeframe of island accumulation, with its three modes of discrete formation, continual process or discrete phases is not well linked to Figure 1b. How do we see these different modes in the Figure? Can examples be given?

We have now improved the link between this text and Figure 1b and included examples. The different modes can be seen in Figure 1b in the start and end point of accumulation, and length of line representing the oldest and youngest radiometric determinations on each island. For example, Warraber has been shown to have formed incrementally over the past 5,500 years beginning approximately 6,000 years ago. In contrast, Dhakandhoo in the Maldives is shown to have formed over a much more discrete timeframe (~1,000 years) with the major phase of island construction being complete 4,000 years ago.

89-90: the text on community experience and adaptation seems speculative. How does the arrival of particular peoples in particular locations relate to particular modes of island development and change? Can the anthropologists supply examples? And, specifically, how is this relevant to Kandahalagalea?

Our argument at this position in the manuscript is not to engage with models of human migration and settlement on islands. Rather, where islands have been shown to have persisted and incrementally or episodically expanded on reef surfaces, and it is known the islands have been inhabited for at least two millennia, these communities have adapted to long-term changes in islands on the reef surfaces. We have amended the sentence to clarify our intent.

104-106: the text anticipates the results here

We have reworded the text to remedy this.

138-142: it seems odd that there is no locational map for these two transects and that it has been hived off to Supplementary Figure 2A.

This decision was made to save space and not clutter an already complex figure. As noted the location of profiles is shown in Supplementary Figure 2.

195: so a 4% difference from 1969 area. So to all intents and purposes = stable?

While the area of the island might be considered near stable, in fact it has expanded in area. More importantly, the island has been anything but static in location on the reef platform. This raises an interesting issue of nomenclature that we are currently elaborating on in a follow-up manuscript. Islands may not change in planform area over decades, but they can migrate 100s of metres on the reef platform, and being precise about the reference to positional stability or stability in island size becomes important for clarity.

205-207: what are the boundaries between erosion / stable / accretion – what are the defining rates?

We provide clear definition of the approach used to detect erosion, accretion and no detectable change in the methods on lines 500-502. A confidence interval of 2σ (95.5%) was applied when calculating shoreline change rates. Transects with statistically significant rates of change are considered erosional (-/ve EPR) or accretionary (+/ve EPR), the remaining transects are classified as exhibiting no detectable change.

260: so are we seeing the 'discrete phases' model of island accumulation? Needs stronger relation to text around lines 85-88

We have strengthened the text around discrete phases of island deposition in the introduction as suggested by the Reviewer.

343: what are these 'nature-based solutions'?

We have modified our treatment of nature-based solutions at the end of the manuscript. However, our point here is that any nature-based solutions should be formulated in full knowledge of the process environment of an island including the rate and trajectories of physical change. Of note, the IPCC AR6 WG2 Chapter on Small Islands spends time on ecosystem-based adaptation (Eba) approaches as part of Nature-Based Solutions. Their Ebas included mangrove replanting, reef rehabilitation beach replenishment traditional tree use to reduce erosion etc. but ignore shoreline and landform change and its implications for adaptation. We have reworded lines 359-363 to reflect these ideas. We would argue of course that that any adaptation solutions should take account of such matters but given the increasing focus on nature based solutions this is an important and timely point to make.

Reviewer #3 (Remarks to the Author):

- The paper provides a tidy and detailed study of the evolution of a Maldivian low reef island. It uses*
- *the work is of significance to the geo-, bio- and social sciences. It is an important contribution to inform on island persistence under global change and so also has political relevance (esp. for small-island states).*
 - *the work supports the conclusions and claims. No further analysis is needed.*
 - *there are no flaws in the data analysis, interpretation and conclusions.*
 - *the methodology is sound*
 - *there is enough detail provided in the methods for the work to be reproduced.*

The text is concise, short yet long enough to provide all needed information. It is very polished and I have no issues here at all. I can clearly follow the arguments.

We thank Reviewer 3 for their positive comments on the manuscript

Graphics are high-quality and suitable for the publication. They should not be shortened.

Thank you

I found only 2 spelling mistakes:

line 137: change "millenia" to "millenium". I think the authors want to say "in the last 1000 years", so the singular, millenium, needs to be used. Millenia is the plural and would mean "in the last several thousand years".

line 170: change to "...two beachrock outcrops", deleting the plural s at the end of beachrocks.

We have amended these issues as identified by the Reviewer.

Some more information would be welcome in the methods:

- radiometric dating: please provide more detail on the actual method, if the allowed space permits.

We have added additional details on the radiometric dating in the methods including the laboratory of analysis and calibration of samples.

- satellite imagery: please provide some more information on what type of imagery (Maxar and Airbus are just the providers), i.e. from which sensor (GeoEye? Sentinel? what pixel resolution?). It would also be interesting to know how that 1969 aerial imagery was geo-referenced to the new images (which come geo-referenced). Using the same method as described for the satellite images?

The satellite imagery used in this study was captured by commercial satellites operated by Maxar and Airbus. These companies are the operators of the satellites used to collect multispectral imagery used in this study. Sentinel is not a commercial satellite operated by either Maxar or Airbus. The images sourced from Airbus were captured by the Pléiades satellites (50cm resolution). The images sourced from Maxar were captured by the QuickBird-2 satellite (Jan. 2005, 60cm resolution), GeoEye-1 (Mar. 2009, 40cm resolution) and WorldView-2 satellite (all other images, 50cm resolution). See table below for summary.

The aerial photo was georeferenced using the same approach for the georeferencing of the satellite imagery as described on lines 436-438.

We have incorporated the above information into the amended methods section.

Date on the figure	Date	Source	Resolution
Jan-69	May-69	Aerial	250cm

Jan-05	10-Jan-05	QuickBird-2	60cm
Mar-09	25-Mar-09	GeoEye-1	40cm
Mar-11	30-Mar-11	WorldView-2	50cm
Jan-14	13-Jan-14	Pleiades	50cm
May-16	5-May-16	Pleiades	50cm
Nov-17	14-Nov-17	WorldView-2	50cm
Feb-18	28-Feb-18	WorldView-2	50cm
Jan-19	26-Jan-19	WorldView-2	50cm
Jan-20	16-Jan-20	Pleiades	50cm
Jan-21	18-Jan-21	WorldView-2	50cm
Oct-21	13-Oct-21	WorldView-2	50cm

- Shoreline calculation. I can follow what was done in the software. But the shoreline changes with tidal state and images are acquired at different tidal states. How was that corrected?

On lines 439-441 we note the edge of island vegetation is used as the shoreline proxy. This is common practice throughout the reef island literature, where instantaneous water line (IWL) derived shorelines are rarely used in multi-decadal scale studies. The edge of vegetation shoreline proxy acts to filter out the short-term “noise” present in the IWL shoreline which is a function of wave runup, tidal level and changes to sea level resulting from quasi-cyclical variability in sea level (i.e. ENSO, PDO). No tidal correction is needed for an edge of vegetation shoreline proxy.

We have not amended the text in response to this comment.

REVIEWERS' COMMENTS

Reviewer #1 (Remarks to the Author):

My original comments on this manuscript were minor and I feel they have been addressed more than sufficiently. I have nothing else to add of great value. I believe this manuscript is ready for publication. Congratulations on a great piece of work!

Reviewer #2 (Remarks to the Author):

The authors have done a very thorough job in addressing the queries raised by the reviewers. In particular, the choice of field site and how the field site text relates to the broader text is much, much clearer. Good. This is, in my view, very close to 'accept'.

A few comments:

As this paper is really about dynamics over different timescales, then the different timescales need to be consistently identified throughout the manuscript. There are some definitions of timescales at lines 123-124 (historical (being split into millennial and centennial) v. recent (last 50 yrs) but it would be helpful if these definitions occurred earlier in the paper. Some text needs to make clearer what is meant by:

Line 54: 'longer term context'

55: 'medium to long term dynamics'

74: 'more distant past'

102: 'short to medium-term dynamism'

106: 'medium-term dynamism'

109: 'shorter term' / 'short term'

201: 'multi-decadal'. First appearance I think. Isn't it just 'historical' in the case of this paper's terminology

Line 323 IODP but line 394 IOD

Line 360: 'renewed enthusiasm for creating nature-based solutions'. So when was there an earlier phase of enthusiasm?

Reviewer #3 (Remarks to the Author):

I have read the responses to reviewers and the reworked version of the manuscript. I feel that all remarks (not only my own, but also those of the other reviewers) have been sufficiently taken care of. This is a polished and informative manuscript that can be accepted now.

RESPONSE TO REVIEWERS' COMMENTS

Reviewer #1 (Remarks to the Author):

My original comments on this manuscript were minor and I feel they have been addressed more than sufficiently. I have nothing else to add of great value. I believe this manuscript is ready for publication. Congratulations on a great piece of work!

We thank the reviewer for their positive comments.

Reviewer #2 (Remarks to the Author):

The authors have done a very thorough job in addressing the queries raised by the reviewers. In particular, the choice of field site and how the field site text relates to the broader text is much, much clearer. Good. This is, in my view, very close to 'accept'.

We thank the reviewer for this positive comment.

A few comments:

As this paper is really about dynamics over different timescales, then the different timescales need to be consistently identified throughout the manuscript. There are some definitions of timescales at lines 123-124 (historical (being split into millennial and centennial) v. recent (last 50 yrs) but it would be helpful if these definitions occurred earlier in the paper. Some text needs to make clearer what is meant by:

We thank the reviewer for highlighting this important issue and we agree that the manuscript should make the distinction between different timescales clear throughout the manuscript.

Line 54: 'longer term context' We have qualified this timescale in the bracketed definition (*millennial-scale*)

55: 'medium to long term dynamics' The sentence now reads.
'.....*understanding medium (centennial) to long-term (millennial) island dynamics.....*'

74: 'more distant past' We have amended the sentence to read: '*.....more distant past (centennial-millennial scales).*'

102: 'short to medium-term dynamism'. Bracketed terms have been used to qualify timescales. Sentence now reads
'.....*obscures any short (decadal) to medium-term (centennial) dynamism in island size.....*'

106: 'medium-term dynamism'. Added '*centennial*' in brackets following 'medium-term'.

109: 'shorter term' / 'short term' We have clarified this as decadal in timescale.

201: 'multi-decadal'. First appearance I think. Isn't it just 'historical' in the case of this paper's terminology. We have defined short-term as decadal earlier in the manuscript, Consequently we think this sub-heading is justified without further qualification.

Line 323 IODP but line 394 IOD. We have amended instances of IODP to IOD.

Line 360: 'renewed enthusiasm for creating nature-based solutions'. So when was there an earlier phase of enthusiasm? We have amended the sentence to read '*...while there has been recent enthusiasm....*'

Reviewer #3 (Remarks to the Author):

I have read the responses to reviewers and the reworked version of the manuscript. I feel that all remarks (not only my own, but also those of the other reviewers) have been sufficiently taken care of. This is a polished and informative manuscript that can be accepted now. We thank the reviewer for the positive comments.